

# Preliminary observations on the age and growth of dog snapper (*Lutjanus jocu*) and mahogany snapper (*Lutjanus mahogoni*) from the Southeastern U.S.

Jennifer C. Potts and  Michael L. Burton

NOAA National Marine Fisheries Service, Beaufort, NC, United States

## ABSTRACT

Dog snapper (*Lutjanus jocu* Bloch and Schneider 1801) and mahogany snapper (*Lutjanus mahogoni* Cuvier 1828) are infrequently caught snappers in the southeastern U.S. primarily occurring off of southern Florida. The species were opportunistically sampled from commercial and recreational fisheries in the southeastern U.S. from 1979 to 2015. Fish were aged (56 dog snapper and 54 mahogany snapper) by counting opaque zones on sectioned sagittal otoliths. Otoliths of both species were easily interpretable and agreement between readers was acceptable. Analysis of otolith edge-type revealed that annuli formed between May and July on both species. Dog snapper ranged from 200–837 mm total length (TL) and ages 2–33, while mahogany snapper ranged from 270–416 mm TL and ages 2–18. The Von Bertalanffy growth equations were $L_t = 746(1-e^{(-0.20(t-0.32))})$ and $L_t = 334(1-e^{(0.31(t+1.19))})$ for dog snapper and mahogany snapper, respectively. The weight-length relations were $W = 1.31 \times 10^{-5} L^{3.03} (n = 78, r^2 = 0.99)$ and $W = 5.40 \times 10^{-6} L^{3.15} (n = 79, r^2 = 0.79)$ for dog snapper and mahogany snapper, respectively, where $W =$ whole weight in grams.

## INTRODUCTION

The dog snapper, *Lutjanus jocu* (Bloch & Schneider, 1801), is a moderate- to large- sized snapper (Lutjanidae) occurring infrequently in commercial and recreational reef fish catches in the southeastern U.S. (SEUS), which includes North Carolina south through the Florida Keys. The species is distributed in the western Atlantic Ocean from North Carolina to Brazil, into the Gulf of Mexico and throughout the Caribbean Sea, though they are rare north of Florida (*Carpenter, 2002*). Dog snapper have also been reported from the eastern Atlantic at Ascension Island (*Lubbock, 1980*) and from the Mediterranean Sea (*Vacchi et al., 2010*). The mahogany snapper, *Lutjanus mahogoni* (Cuvier, 1828), a smaller member of the Lutjanidae also caught infrequently by fishers, is distributed in the western Atlantic Ocean from the Carolinas to Venezuela and throughout the Caribbean Sea, including the Gulf of Mexico (*Carpenter, 2002*). Both species are found on coral reefs and rocky hardbottom habitat at depths up to 100 m. Mahogany snapper are known to form large

Corresponding author
Jennifer C. Potts,
jennifer.potts@noaa.gov

social aggregations (*McEachran & Fechhelm, 2005*), while dog snapper are typically solitary and wary except when in spawning aggregations (*Domeier, Koenig & Coleman, 1996*).

Both species are of limited importance to the SEUS reef fish fishery, and the sparse estimated landings for both species reflect their low occurrence in the SEUS region (Table 1). Also, these less frequently landed snappers tend to be lumped into a "unclassified snappers" category in recreational and commercial landings reporting, thus making more definitive species-specific landings discrimination difficult. Few or zero intercepts of the species by port agents did not allow the two separate recreational surveys, Southeast Region Headboat Survey (SRHS) and Marine Recreational Intercept Program (MRIP), to estimate landings by weight, thus landings were only recorded by number of fish landed. In the SEUS, recreational fisheries landings were estimated at an average of 1236 dog snapper and 1122 mahogany snapper per year from 1981–2015 (*NMFS, 2016a*). On the other hand, commercial landings were reported by weight, not numbers of fish. Average commercial landings were 490 kg per year from 1991–2014 for *L. jocu*, while commercial landings of *L. mahogoni* were virtually non-existent (79 kg total from 1986-2014) (*NMFS, 2016b*). Dog snapper landings in the U.S. Caribbean (Puerto Rico and the U.S. Virgin Islands –USVI) were higher, with recreational landings averaging 7227 fish annually between 2000–2015 (*NMFS, 2016a*) and commercial landings averaging 120 kg per year from 1988–2012 (*NMFS, 2016b*). Recreational landings of mahogany snapper from the U.S. Caribbean averaged 669 fish annually from 2000–2015 (*NMFS, 2016a*) while commercial landings averaged 173 kg per year from 1988–2012 (D Gloeckner, pers. comm., 2015).

Published studies on the life history of *L. jocu* are limited to age and growth studies from Brazil (*Rezende & Ferreira, 2004*; *Previero et al., 2011*) and Cuba (*Claro, Sierra & Garcia-Arteaga, 1999*), the documentation of spawning aggregations from Belize (*Carter & Perrine, 1994*; *Heyman et al., 2001*) and the USVI (*Kadison et al., 2006*; *Biggs & Nemeth, 2016*), and habitat use by juveniles through adults on the Abrolhos Shelf of Brazil (*Moura et al., 2011*). *Franco & Olavo (2015)* examined commercial fishery landings data from northeastern Brazil as indirect evidence for the presence of spawning aggregations of dog snapper. *Aschenbrenner, Hackradt & Ferreira (2016)* examined habitat selection in early life history stages of dog snapper in northeastern Brazil. Studies of *L. mahogoni* are limited to examinations of the importance of various habitat types in Caribbean bays to juvenile stages (*Nagelkerken et al., 2000a*; *Nagelkerken et al., 2000b*).

We studied these two species from the SEUS because little is known of their life history, which is important to managers interested in multi-species or ecosystem-based management. Both species were managed by the South Atlantic Fishery Management Council (SAFMC) under the Snapper-Grouper Fishery Management Plan (FMP) from 1983 until June 22, 2016. At that time, the SAFMC made the decision to relegate management of these species and several others to individual state resource agencies due to the low magnitude of landings in federal waters. The species are currently managed by the Florida Fish and Wildlife Conservation Commission (FWCC) with a 12-inch (305 mm) total length (TL) size limit and inclusion in a 10 snapper per person daily aggregate bag limit (*FWCC, 2016*). Increasing restrictions on more commonly caught reef fish species will likely lead to increased harvest of less common species such as dog snapper or mahogany

**Table 1 Reported fisheries landings for dog snapper and mahogany snapper from the SEUS and U.S. Caribbean, 1981–2015.**

| Year | Dog snapper | | | | | Mahogany snapper | | | | |
| | SEUS | | | U.S. Caribbean | | SEUS | | | U.S. Caribbean | |
| | SRHS No. | MRIP No. | Comm kg | MRIP No. | Comm kg | SRHS No. | MRIP No. | Comm kg | MRIP No. | Comm kg |
|---|---|---|---|---|---|---|---|---|---|---|
| 1981 | 8 | 3238 | | | | 9 | 8484 | | | |
| 1982 | 7 | | | | | 42 | 2300 | | | |
| 1983 | 21 | | | | | 12 | 824 | | | |
| 1984 | 92 | 1169 | | | | 15 | 17048 | | | |
| 1985 | 2 | | | | | 51 | 4065 | | | |
| 1986 | 103 | 3946 | | | | 15 | | | | |
| 1987 | 25 | | | | | 17 | | | | |
| 1988 | 20 | | | | 31 | 34 | | | | 69 |
| 1989 | 17 | | | | 5 | 6 | | | | 1482 |
| 1990 | 86 | | | | | | | | | |
| 1991 | 166 | | 113 | | 142 | 12 | | | | |
| 1992 | 344 | | 150 | | 44 | 5 | | | | 34 |
| 1993 | 181 | 1600 | 142 | | 270 | 9 | | | | 30 |
| 1994 | 379 | | 115 | | 206 | 4 | | | | 62 |
| 1995 | 265 | | 394 | | 30 | 19 | | | | 233 |
| 1996 | 88 | 1227 | 894 | | 108 | 79 | | 35 | | 133 |
| 1997 | 122 | | 926 | | 6 | 49 | | | | 569 |
| 1998 | 144 | 1304 | 1012 | | | | | 44 | | 160 |
| 1999 | 30 | 925 | 444 | | 45 | 198 | 2462 | | | 25 |
| 2000 | 34 | 1687 | 385 | 7815 | 59 | | | | 2817 | 33 |
| 2001 | 70 | 896 | 880 | 22067 | 1026 | | | | 928 | 5 |
| 2002 | 56 | 318 | 488 | 17258 | 65 | | 643 | | | |
| 2003 | 20 | 1415 | 875 | 4445 | 12 | | | | 2234 | 7 |
| 2004 | 40 | 670 | 766 | 2542 | | | | | 651 | |
| 2005 | 18 | 1895 | 314 | 1035 | | | | | | |
| 2006 | 330 | 648 | 913 | 1771 | 33 | | | | | |
| 2007 | 25 | 14364 | 540 | 18767 | 188 | | 1530 | | 704 | |
| 2008 | 130 | | 518 | 4435 | 60 | 43 | | | | |
| 2009 | 11 | 2759 | 518 | 12130 | | 22 | 176 | | 668 | 12 |
| 2010 | 101 | 586 | 164 | 2307 | 10 | 23 | | | 455 | 9 |
| 2011 | 34 | 325 | 330 | 3890 | 34 | | | | | 137 |
| 2012 | 45 | 486 | 345 | 4694 | 29 | | | | | 586 |
| 2013 | 51 | | 296 | 4798 | | 116 | | | | 718 |
| 2014 | 24 | 28 | 249 | | | 176 | | | | 537 |
| 2015 | 65 | 627 | | 7679 | | 184 | 595 | | 2243 | 2 |
| Σ | 3154 | 40113 | 11771 | 115633 | 2403 | 1140 | 38127 | 79 | 10700 | 4843 |

**Notes.**

SRHS, Southeast Region Headboat Survey; MRIP, Marine Recreational Intercept Program; Comm, Commercial statistics.

snapper. Managers need to understand the growth of a species, since that can be used to estimate its reproductive potential and mortality rate. To that end, we used sectioned otoliths to determine ages of dog snapper and mahogany snapper from the SEUS and to estimate seasonality of annulus formation. We also derived theoretical growth parameters and determined length-length and weight-length relationships.

## METHODS

### Age estimation and timing of opaque zone formation

Dog snapper and mahogany snapper were opportunistically sampled from fisheries operating offshore of North Carolina through Key West, FL. All specimens used in this study were killed as part of legal fishing operations and were already dead when sampled by the port agents; thus all research was conducted in accordance with the Animal Welfare Act (AWA) and with the US Government Principles for the Utilization and Care of Vertebrate Animals Used in Testing, Research, and Training (USGP) OSTP CFR, May 20, 1985, Vol. 50, No. 97. All fish were captured by either conventional vertical hook and line gear or by spears. Sagittal otoliths were collected from 62 dog snapper and 57 mahogany snapper by National Marine Fisheries Service port agents sampling the recreational headboat and commercial fisheries from 1979 to 2015. Total (TL) and fork lengths (FL) of specimens were recorded in millimeters (mm), and whole weight was recorded in grams (g) for fish sampled by the Southeast Region Headboat Survey (SRHS). Weights were generally unavailable for fish landed by the commercial fisheries, as these fish were eviscerated at sea. Sagittal otoliths were removed through the otic bulla inside the gill cavity and stored dry in coin envelopes. Otoliths were sectioned on a low-speed saw, following the methods of *Potts & Manooch III (1995)*. Three serial 0.5 mm sections were taken, with at least one of them encompassing the otolith core. The sections were mounted on microscope slides with thermal cement and covered with histological mounting medium before analysis. The sections were viewed under a dissecting microscope at 12.5X using reflected light (Fig. 1). Each sample was assigned an opaque zone count by two readers. Counts were compared between readers. An index of average percent error (APE) was calculated, following the methodology of *Beamish & Fournier (1981)*. Where two readings for a specimen disagreed, the sections were viewed again. If agreement was reached the sample was retained; otherwise, the sample was discarded from further analysis.

We assessed opaque zone periodicity using otolith edge, or margin, analysis by visual categorization. The edge type of the otolith was noted: 1 = opaque zone forming on the edge of the otolith section; 2 = narrow translucent zone on the edge, generally less than 30% of the previous translucent zone; 3 = moderate translucent zone on the edge, generally 30% to 60% of the previous translucent zone; 4 = wide translucent zone on the edge, generally greater than 60% of the previous translucent zone (*Harris et al., 2007*). All samples were assigned an age based on edge frequency analysis metrics, opaque zone count and time of capture. The zone count was increased by one, to reflect the calendar age of the fish, if the specimen was caught before increment formation and had an edge that was a moderate to wide translucent zone (type 3 or 4).

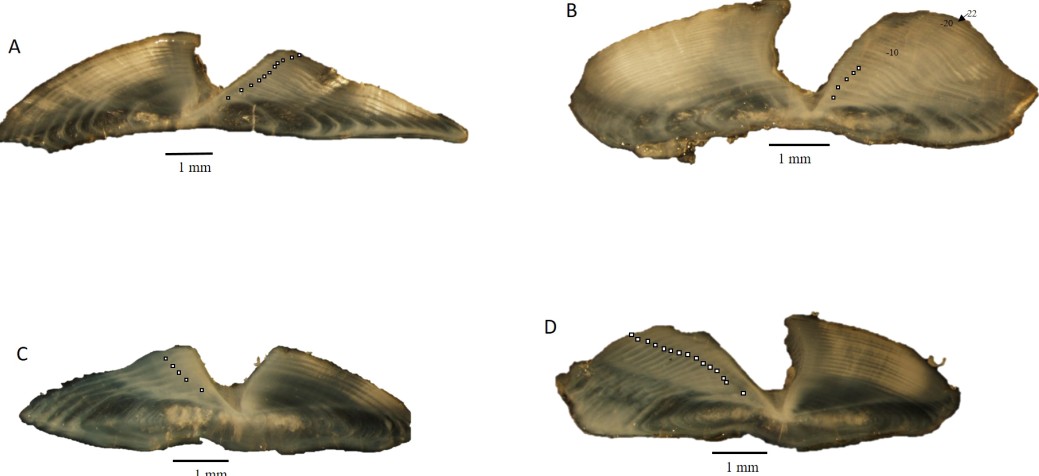

**Figure 1** Photographs of transverse sections of sagittal otoliths from (A). dog snapper (*Lutjanus jocu*), age 11, edge type 1–687 mm TL, captured 07/03/2015; (B) dog snapper (*Lutjanus jocu*), age 22, edge type 4–708 mm TL, captured 07/03/2012; (C) mahogany snapper (*Lutjanus mahogoni*), age 5, edge type 4–332 mm TL, captured 04/13/1999; (D) mahogany snapper (*Lutjanus mahogoni*), age 14, edge type 1–330 mm TL, captured 07/01/2005.

## Theoretical growth

*Von Bertalanffy (1938)* growth parameters were estimated from the observed length-at-age data. Calendar age was used since little information on reproduction of these species in U.S. waters was available to assign a biological age. Growth parameters were derived initially by minimizing the negative sum of log-likelihoods with the AD Model Builder estimation software (Otter Research Ltd., Sidney, B.C., Canada). We examined parameters computed from a freely-estimated model, assuming normal distribution of lengths about each calendar age, and a model that adjusted for size selectivity bias within the fishery. Because samples for this study were obtained from fishery landings, the estimate of growth at the youngest ages may be skewed due to minimum size regulations imposed on the fishery or selection of the fish by the fishers. As an alternate strategy to model growth, the von Bertalanffy growth parameters were estimated using a left-truncated normal probability density function on length for fish subjected to the minimum size limit (305 mm TL) regulation, as developed by *McGarvey & Fowler (2002)*. For samples in this study not subject to minimum size limit, the full, untruncated normal likelihood was used.

## Body size relationships

Important metrics of fish species include the relationship between weight and length and between various length measurements. Whole fish weight (g) was regressed on TL using data for all dog snapper and mahogany snapper measured by the SRHS from 1979–2015 ($n = 78$, *L. jocu*; $n = 79$, *L. mahogoni*). We evaluated both direct fits using nonlinear least squares regression (SAS Institute, Cary, NC, USA) and a linearized fit of the log-transformed data, examining the residuals to determine which fit was more appropriate. We used the same SRHS data to examine the relationships between TL and FL using linear regression ($n = 48$, dog snapper; $n = 65$, mahogany snapper).

**Table 2** Number of otolith samples available for age-growth study of dog snapper and mahogany snapper from the Southeastern U.S.

| Fishery | Florida | N. Carolina | S. Carolina | Total |
|---|---|---|---|---|
| Dog Snapper | | | | |
| Commercial | 26 | 5 | 1 | 32 |
| Recreational | 24 | – | – | 24 |
| Total | 50 | 5 | 1 | 56 |
| Mahogany Snapper | | | | |
| Commercial | 10 | – | – | 10 |
| Recreational | 44 | – | – | 44 |
| Total | 54 | – | – | 54 |

## RESULTS

### Age estimation

A total of 56 sagittal otoliths from *L. jocu* were sectioned (six otoliths were broken in storage). Fifty-four otoliths from *L. mahogoni* were sectioned, (three were broken in storage). The distribution by fishery and state of age samples is shown in Table 2. Dog snapper samples were primarily from southeastern Florida and Florida Keys waters (25 from commercial fishery landings and 25 from headboat angler landings) but did include five samples from the North Carolina commercial fishery and one sample from the South Carolina commercial fishery. All mahogany snapper came from southeastern Florida and Florida Keys waters, with 80% coming from recreational headboat fisheries and 20% coming from commercial landings.

Sectioned otoliths for both species were clear and easy to interpret (Fig. 1). Opaque zones were counted on all sectioned otoliths. Initial readings by the two authors resulted in 61% and 69% agreement for dog snapper and mahogany snapper, respectively. When we used ±1 year, agreement increased to 95% and 98% respectively. APE was 3.29% for dog snapper and 1.80% for mahogany snapper. These were both well within the criteria of 5% determined acceptable by *Campana (2001)*. The age readers reviewed the samples where there was disagreement on the age and were able to reach consensus. Thus, all age data were used in subsequent analysis.

### Timing of opaque zone formation

Opaque zones on the otolith marginal edge occurred May through July for both species (Fig. 2). Otoliths from dog snapper were without an opaque zone on the edge from August through April, while a single mahogany snapper caught in November exhibited an opaque edge. Dog snapper exhibited the least amount of translucent edge from August through October, with the width of the translucent edge increasing until reaching a maximum January through April, prior to opaque zone deposition in May. Mahogany snapper exhibited a similar pattern. We concluded that opaque zones on otoliths of both species were annuli.

We assigned calendar, or chronological, ages as follows: for fish caught January through July and having an edge type of 3 or 4, the annuli count was increased by one; for fish

a.

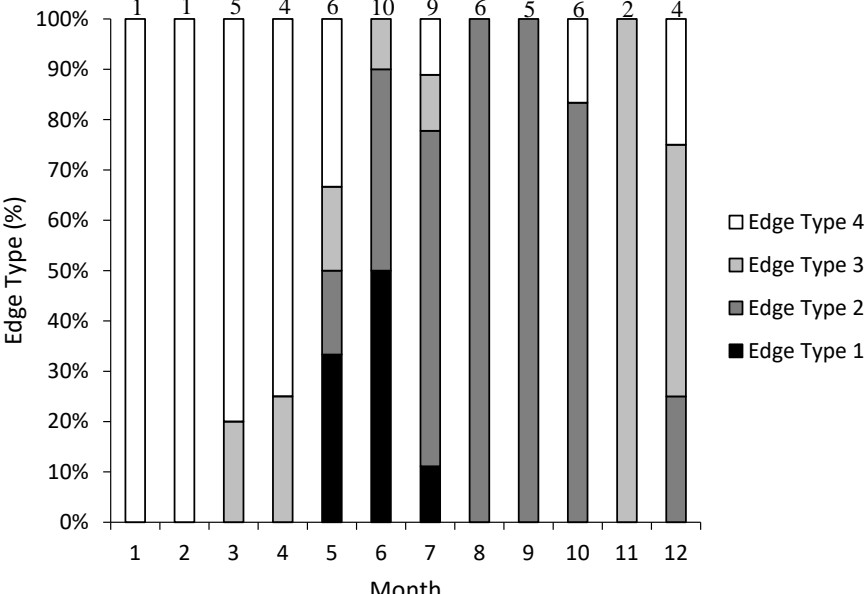

b.

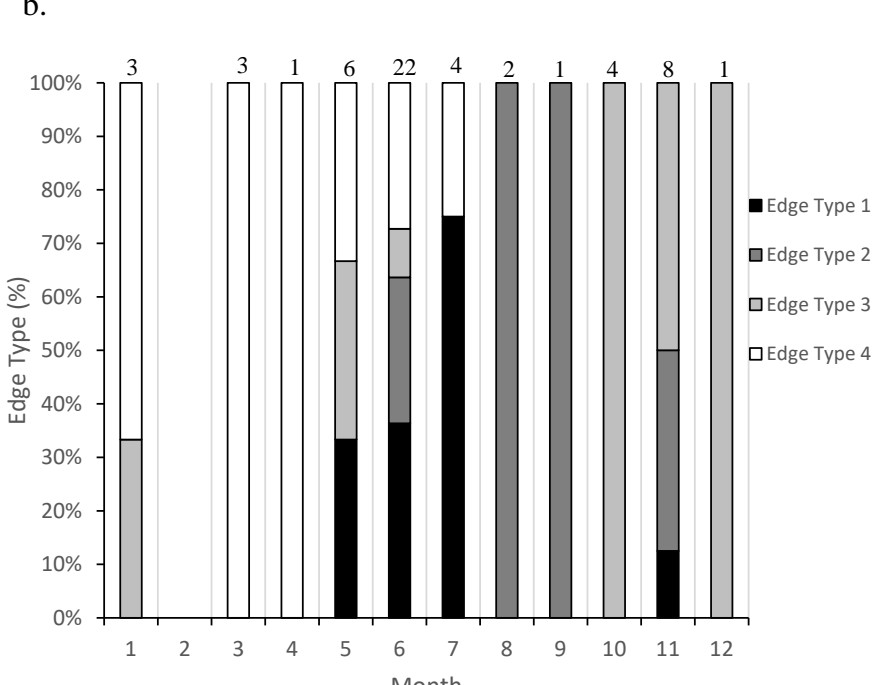

**Figure 2** Frequency of edge type by month (marginal increment analysis) of (A) dog snapper (*Lutjanus jocu*) and (B) mahogany snapper (*Lutjanus mahogoni*) otoliths from the SEUS: edge type 1 = opaque zone on edge; edge type 2 = narrow translucent zone on edge (<30% of previous translucent zone); edge type 3 = moderate translucent zone on edge (30%–60% of previous translucent zone); and edge type 4 = wide translucent zone on the edge (>60% of previous translucent zone). Numbers above each bar were the sample size for each month.

caught in that same time period with an edge type of 1 or 2 and for fish caught from August to December, the calendar age was equivalent to the annuli count.

## Growth

Dog snapper in this study ranged from 200–836 mm TL and from ages 2–33, but only four fish were older than age-12 (Table 3). While 50 out of 57 fish (88%) were from Florida, the largest and oldest fish was from South Carolina. Predicted size-at-age of dog snapper agreed well with mean observed size-at-age (Fig. 3A). The Von Bertalanffy parameters (±1 Std. Err.) are presented in Table 4. The growth model parameters estimated when correcting for the left-truncated normal distribution imposed by the minimum size limit regulation (*McGarvey & Fowler, 2002*) are also in Table 4. The model appeared to have a more realistic estimate of initial growth (Fig. 3A; Table 4). Comparing negative log likelihood values from each model, the model incorporating the correction for the truncated normal distribution due to the size limit regulation was a better fit to the data than when assuming no bias (-log likelihood: 294.65 v. 343.95).

Mahogany snapper ranged from 270–416 mm and ages 2–18. All samples were from south Florida waters. Predicted size-at-age for the freely estimated model run agreed reasonably well with mean observed size-at-age (Fig. 3B), and the von Bertalanffy parameters (±1 Std. Err.) are presented in Table 4. As with dog snapper, the mahogany snapper specimens available for this study lacked smaller fish due to the fishery-dependent nature of the samples; the resulting growth parameters using the method of *McGarvey & Fowler (2002)* are in Table 4 and graphically presented in Fig. 3B. As with dog snapper, the model assuming the truncated normal distribution on lengths at the youngest ages was a better fit to the data than the freely estimated model (-log likelihood: 222.85 v. 245.57).

## Weight–Length relations

When performing the statistical analyses of $W$–$L$ relations, a multiplicative error term (variance increasing with size) in the residuals was revealed for both dog and mahogany snappers. A linearized ln-transform fit of the data was appropriate, and the resulting equations were:

Dog snapper—$Ln(W) = 3.028 \ln(L) - 11.249; (n = 78, r^2 = 0.99)$

Mahogany snapper—$Ln(W) = 3.154 \ln(L) - 12.136; (n = 79, r^2 = 0.79)$

where $W$ = whole weight in grams and $L$ = total length in mm. The equations were transformed back to the power equation form, $W = a L^b$, after adjusting the intercepts for log-transformation bias with the addition of one-half of the mean-square error (1/2 MSE) (*Beauchamp & Olson, 1973*). The resulting regression equations were:

Dog snapper—$W = 1.31 \times 10^{-5} L^{3.03} (n = 78, MSE = 0.013)$ (Fig. 4A) and

Mahogany snapper—$W = 5.40 \times 10^{-6} L^{3.15} (n = 78, MSE = 0.014)$ (Fig. 4B).

**Table 3** Observed mean and predicted total length from size-limit corrected growth model (TL, mm) of dog snapper (*Lutjanus locu*) and mahogany snapper (*Lutjanus mahogoni*) collected from 1979–2015 along the southeastern U.S. coast.

| | | Dog snapper | | | | | Mahogany snapper | | |
|---|---|---|---|---|---|---|---|---|---|
| Age | n | Mean TL (±SE) | TL range | Pred. TL | Age | n | Mean TL (±SE) | TL range | Pred. TL |
| 2 | 6 | 298 (24) | 200–351 | 213 | 2 | 1 | 300 | | 210 |
| 3 | 7 | 322 (22) | 258–401 | 310 | 3 | 1 | 270 | | 243 |
| 4 | 8 | 384 (28) | 267–480 | 389 | 4 | 1 | 325 | | 267 |
| 5 | 7 | 491 (22) | 376–522 | 453 | 5 | 3 | 308 (13) | 285–330 | 285 |
| 6 | 6 | 505 (36) | 334–575 | 506 | 6 | 5 | 331 (7) | 304–347 | 298 |
| 7 | 6 | 585 (35) | 423–655 | 550 | 7 | 5 | 330 (7) | 307–346 | 308 |
| 8 | 2 | 483 (143) | 340–626 | 585 | 8 | 5 | 335 (6) | 318–355 | 315 |
| 9 | 6 | 626 (31) | 518–733 | 615 | 9 | 5 | 342 (8) | 322–367 | 320 |
| 10 | 2 | 704 (17) | 687–720 | 638 | 10 | 4 | 323 (7) | 305–336 | 324 |
| 12 | 2 | 510 (189) | 322–699 | 674 | 11 | 5 | 350 (12) | 320–380 | 326 |
| 13 | 1 | 755 | | 687 | 12 | 5 | 339 (20) | 303–416 | 328 |
| 21 | 1 | 708 | | 734 | 13 | 4 | 337 (20) | 307–359 | 330 |
| 22 | 1 | 763 | | 736 | 14 | 5 | 360 (14) | 330–407 | 331 |
| 33 | 1 | 837 | | 745 | 15 | 2 | 341 (0.5) | 341–342 | 332 |
| | | | | | 16 | 2 | 330 (10) | 320–340 | 332 |
| | | | | | 18 | 1 | 322 | | 333 |

**Table 4** Von Bertalanffy parameters and associated statistics for dog snapper (*Lutjanus jocu*) and Mahogany snapper (*Lutjanus mahogoni*) collected from the southeastern U.S. from 1979–2015, with comparison to parameters from studies of dog snapper from other areas.

| Study/Parameter | $L_\infty$ | SE | K | SE | $t_0$ | SE |
|---|---|---|---|---|---|---|
| Dog snapper—this study, freely estimated | 783 TL | 75 | 0.15 | 0.05 | −1.30 | 1.09 |
| Dog snapper—this study, bias-corrected | 746 TL | 78 | 0.20 | 0.09 | 0.32 | 1.44 |
| Dog snapper—Cuba (*Claro, Sierra & Garcia-Arteaga, 1999*) | 854 FL (903 TL) | | 0.10 | | −2.00 | |
| Dog Snapper—Brazil (*Rezende & Ferreira, 2004*) | 772 FL (817 TL) | | 0.11 | | −3.73 | |
| Dog snapper—Brazil (*Previero et al., 2011*) | 878 FL (928 TL) | | 0.10 | | −1.49 | |
| Mahogany snapper—this study, freely estimated | 346 TL | 7 | 0.28 | 0.13 | −4.18 | 3.42 |
| Mahogany snapper—this study, bias-corrected | 334 TL | 17 | 0.31 | 0.32 | −1.19 | 6.71 |

## Length–Length relations

The relationships between TL and FL as determined by linear regression are described by the following equations:

Dog snapper : $\text{TL} = 1.05\,\text{FL} + 6.40$ $(n = 48, r^2 = 0.99)$

$\text{FL} = 0.95\,\text{TL} - 5.17$ $(n = 48, r^2 = 0.99)$

Mahogany snapper : $\text{TL} = 1.03\,\text{FL} + 9.27$ $(n = 65, r^2 = 0.93)$

$\text{FL} = 0.89\,\text{TL} + 12.01$ $(n = 65, r^2 = 0.93)$.

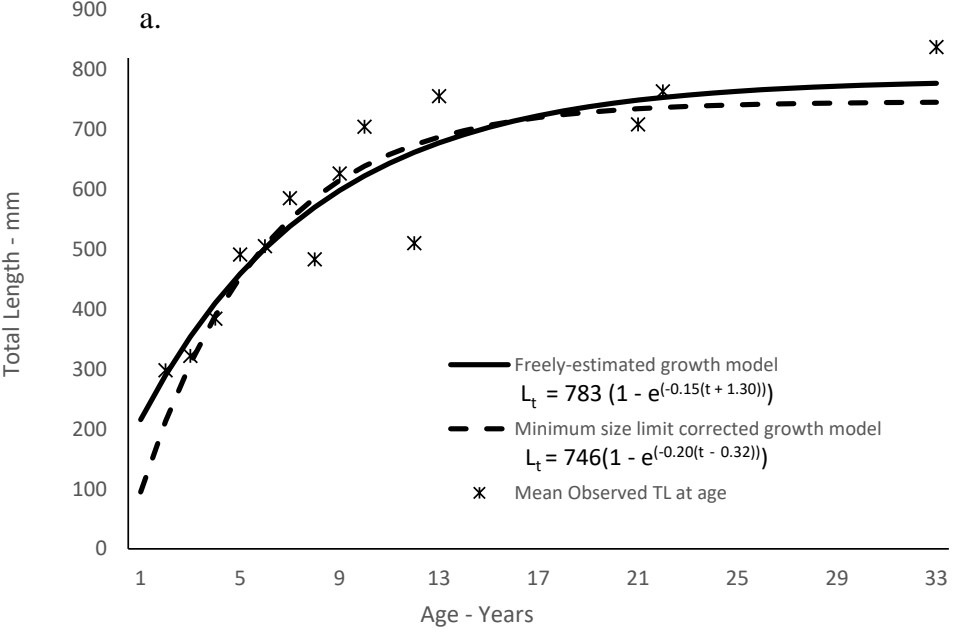

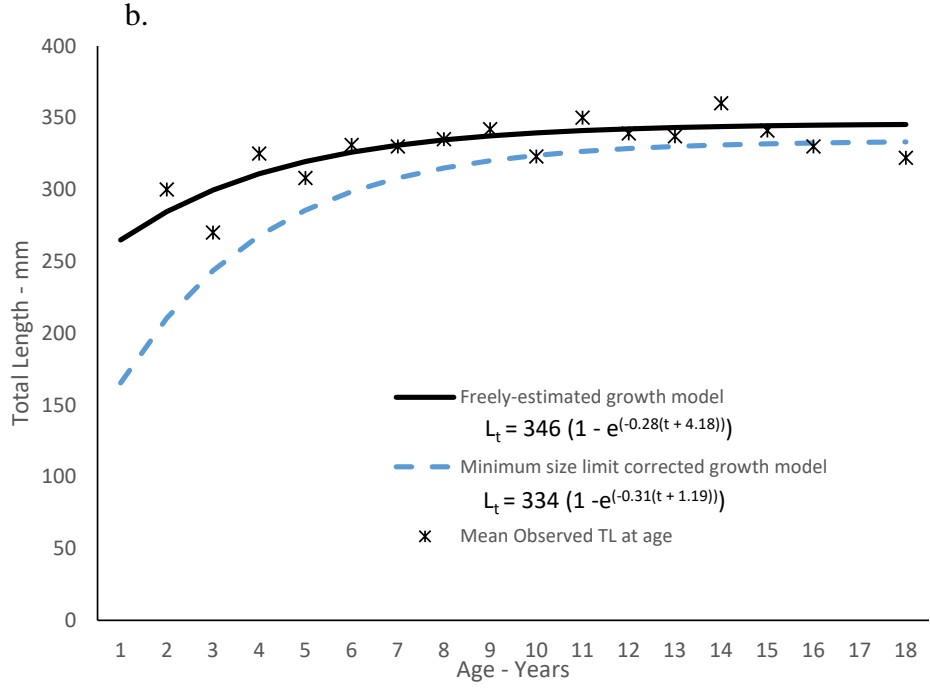

**Figure 3** Observed and predicted lengths-at-age for (A) dog snapper (*Lutjanus jocu*) and (B) mahogany snapper (*Lutjanus mahogoni*), sampled from the southeastern United States from 1979–2015, measured in total lengths (TL).

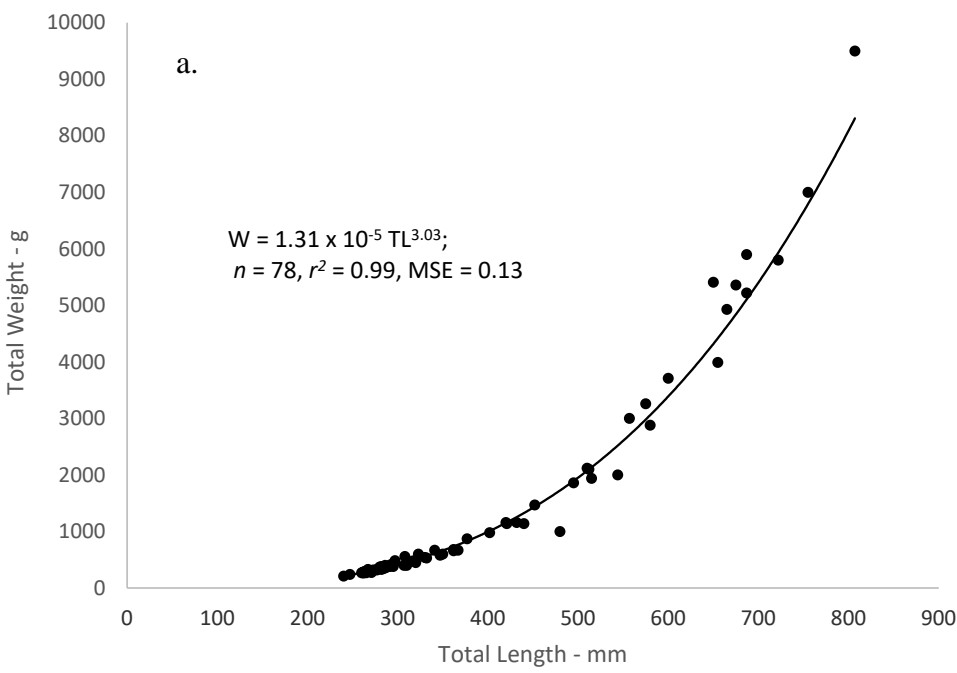

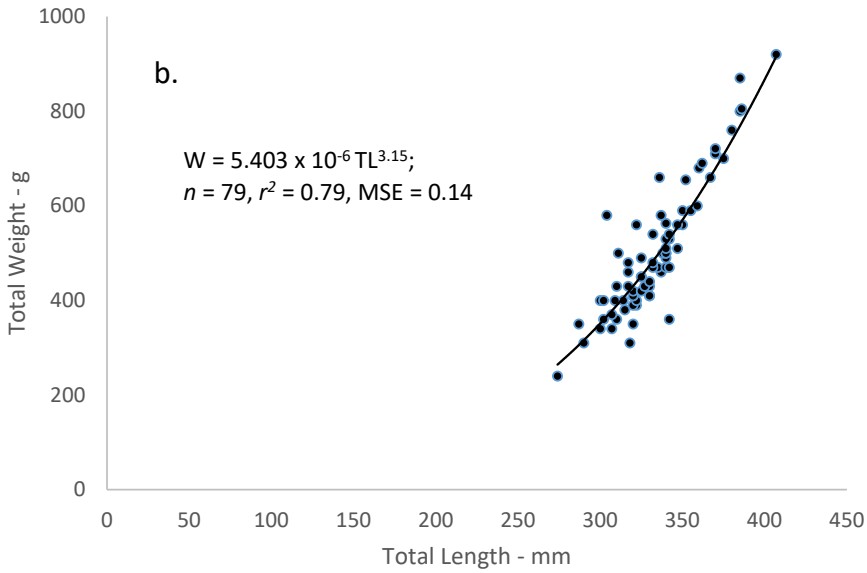

**Figure 4** Scatter plot of whole weight—total length relationship for (A) dog snapper (*Lutjanus jocu*) and (B) mahogany snapper (*Lutjanus mahogoni*) sampled from the southeastern United States from 1979–2015.

## DISCUSSION

Opaque zones in the otoliths of both dog snapper and mahogany snapper were determined to be deposited once per year between May and July. Other SEUS lutjanids display similar timing of annulus deposition. Gray snapper (*Lutjanus griseus* Linnaeus, 1758) deposited annuli in June and July (*Burton, 2001*), and mutton snapper (*Lutjanus analis* Cuvier 1828) deposited annual rings from March to May (*Burton, 2002*). Cubera snapper (*Lutjanus cyanopterus* Cuvier, 1828) were found to deposit annuli from April to August, peaking in May and June (*Burton & Potts, 2017*).

Like many *Lutjanus spp.*, dog snapper grew reasonably fast, attaining an average observed size of 322 mm, 491 mm, 585 mm, 704 mm and 755 mm by ages 3, 5, 7, 10 and 13 respectively (Table 3, Fig. 3A). The predicted growth curve from the freely-estimated growth model fit the observed data well and could be used to describe fish recruited to the fishery. Due to the minimum size limit regulations, the smallest and youngest fish were excluded from the fishery landings, and these regulations have the effect of capturing the fastest growers at the youngest ages retained in the fishery landings. Because of this selectivity, the freely estimated model can represent growth of fish in the fishery, but over-estimated initial growth of the fish in the population. The use of the size limit bias corrected growth model has become standard practice in U.S. Southeast Data, Assessment and Review (SEDAR) stock assessments since 2005 (starting with *SEDAR, 2005*). In the case of yellowmouth grouper (*Mycteroperca interstitialis*) where no fish in the study was younger than age-3, the use of the bias-corrected model yielded a more biologically reasonable growth model (*Burton, Potts & Carr, 2014*). We feel the correction to the dog snapper growth model imposed by the bias in selectivity of the fish in the fishery more accurately estimates the growth of the fish in the population (Fig. 3A). Growth parameters and associated standard errors for both model runs are given in Table 4.

While this study is the first to examine growth of dog snapper from SEUS waters, other studies from the Caribbean found similar parameters (Table 4). *Rezende & Ferreira (2004)* estimated $L_\infty$ close to what we estimated in our study, but the $K$ and $t_0$ values were very different. Because of the much larger negative $t_0$ in their study, initial growth of the fish was not biologically reasonable, which in turn under-estimated $K$. In the case of *Claro, Sierra & Garcia-Arteaga (1999)* and *Previero et al. (2011)*, the $t_0$ values they present approached what we estimated. The $L_\infty$ values were much higher, though, and with the inverse correlation of $L_\infty$ to $K$, the $K$ values from those studies were lower than ours. Red snapper (*L. campechanus*), a close congener and co-occurring species, also show fast initial growth ($K = 0.24$; *SEDAR, 2016*) attaining asymptotic length relatively quickly compared to some other large reef fish species. We feel that more attention should be given to estimating fish growth.

The oldest dog snapper in our study was 33 years old and was caught in the South Carolina commercial fishery. The vast majority of dog snapper samples came from Florida waters, where the oldest fish was 22 years old. The studies by *Rezende & Ferreira (2004)* and *Previero et al. (2011)* demonstrated that the species certainly has a longevity beyond what we found from Florida fish. From the limited data on depth of fishing associated with the

dog snapper samples (15–73 m), our oldest fish was caught in the deepest water recorded (73 m), which supports the findings of *Moura et al. (2011)* that dog snapper perform cross shelf ontogenetic migrations from inshore to offshore as they get older. We would expect the oldest and largest fish to be caught from deeper offshore habitats.

Mahogany snapper are a smaller, shorter-lived fish than dog snapper, and as such exhibit a much smaller length distribution, with maximum TL just over 400 mm. The mean observed length-at-age data fits the growth curve fairly well, but because of lack of age-1 fish and the paucity of samples for all ages below age-5, it does not do a good job of describing the growth of the early part of the growth trajectory. For this reason, we modeled growth using the bias-corrected model of *McGarvey & Fowler (2002)*. This model had little effect on the estimates for the parameters $L_\infty$, $K$ increased slightly, but resulted in a change in the value of $t_0$ from $-4.18$ to a more biologically realistic value of $-1.19$ (Table 4). This value resulted in a theoretical size-at-age for mahogany snapper of 104 mm at age-0, 165 mm at age-1, and 210 mm at age-2.

Though the number of samples for this study were limited, we have seen similar patterns of growth in other species in the genus *Lutjanus* in the SEUS. Dog snapper is one of the larger Lutjanids and exhibits similar fishery growth parameters to mutton snapper, *L. analis* (*Burton, 2002*). The growth coefficients ($K$) of both species, 0.15 for dog snapper and 0.16 for mutton snapper (*Burton, 2002*), indicate that both species take longer time to reach their maximum size, compared to smaller snapper species). The fact that we found a 33 yr old dog snapper within so few age samples suggests that their longevity could approach that of mutton snapper (up to 40 years; *SEDAR, 2015*). The adults of both species have been described as generally solitary and wary except at the time of aggregating to spawn (*Domeier, Koenig & Coleman, 1996*). On the other hand, mahogany snapper is one of the smallest *Lutjanus* species, and more similar to lane snapper, *L. synagris*, in growth and behavior. *Brennan (2004)* estimated the Von Bertalanffy growth parameters for lane snapper from the commercial and recreational fisheries of southeast Florida similar to the age samples we had available for this study. Both mahogany snapper and lane snapper attain their maximum size more rapidly than their larger congeners and do not tend to live as long $-K = 0.31$ and max age $= 18$ years for mahogany snapper and $K = 0.34$ and max age $= 12$ years for lane snapper. Also, these smaller lutjanids form social aggregations during the day. The similarities within groups of lutjanids could allow managers to make predictions on fishery impacts to the less common species based on behavior and growth patterns of more well studied species.

We have demonstrated in this study that sectioned sagittal otoliths are an effective method for the aging of dog snapper and mahogany snapper. We have presented the first description of life history parameters for these two species for samples from SEUS waters and the first description in the literature for mahogany snapper age and growth. While the magnitude of landings from mainland SEUS waters for both species is low, they are of more importance in the U.S. Caribbean and these studies could contribute to effective management in these locales. Another equally important reason for studying species for which we have little information is that eventually this information is likely to be needed

for inclusion in multispecies stock assessments or ecosystem-based assessment models (*Christensen et al., 2009*).

## ACKNOWLEDGEMENTS

We extend thanks to all the NMFS and state port agents who collected otoliths from commercial and headboat fisheries for this study. Thanks are also due to Vivian Matter and David Gloeckner of the Miami NMFS lab for supplying Caribbean landings data, Tom Sminkey from the NMFS Office of Science and Technology, Silver Spring MD for supplying recreational landings data. Thanks to Rob Cheshire and Nate Bacheler for initial reviews of this manuscript, as well as the other anonymous reviewers that made this manuscript stronger.

### Funding

The authors received no funding for this work.

### Competing Interests

The authors declare there are no competing interests.

### Author Contributions

- Jennifer C. Potts and Michael L. Burton conceived and designed the experiments, performed the experiments, analyzed the data, contributed reagents/materials/analysis tools, wrote the paper, prepared figures and/or tables, reviewed drafts of the paper.

### Animal Ethics

The following information was supplied relating to ethical approvals (i.e., approving body and any reference numbers):

All specimens used in this study were killed as part of legal fishing operations and were already dead when sampled by the port agents, thus all research was conducted in accordance with the Animal Welfare Act (AWA) and with the US Government Principles for the Utilization and Care of Vertebrate Animals Used in Testing, Research, and Training (USGP) OSTP CFR, May 20, 1985, Vol. 50, No. 97.

### Data Availability

The raw data for this study are considered confidential data by US Federal Law—Magnuson-Stevens Act -Confidential–50 CFR, Part 600. Title IV, Section 402, subsection b.

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
