# Peer review of "Preliminary observations on the age and growth of dog snapper (Lutjanus jocu) and mahogany snapper (Lutjanus mahogoni) from the Southeastern U.S"

_PeerJ, doi:10.7717/peerj.3167_

## Round 0.1 · original submission · Minor Revisions

Please, amend your manuscript according to the suggestions given.

Reviewer 1 ·

Basic reporting

see attached review

Experimental design

see attached review

Validity of the findings

see attached review

Additional comments

see attached review

Annotated reviews are not available for download in order to protect the identity of reviewers who chose to remain anonymous.

Reviewer 2 ·

Basic reporting

I printed the text and made many comments directly on that copy - which I scanned into the attached pdf, minus the figures. The pdf I received had the figures somewhat poorly aligned. I had, though, no problem in deciding the general layout.

This paper solidly details, as the authors admit, a preliminary effort to document length/weight, age and growth dynamics of these two relatively rarely encountered species within the geographic range studied. The preliminary aspect here is a result of the great length of time during which specimens were available (weak cohort representation), and the relative paucity of specimens observed over the wide age ranges for each species.

Also admitted by the authors, is that this type of information is critical to future potential management regulations for the larger snapper-grouper species complex. It seems then that this paper should be made available to your readership.

Experimental design

This study is a very straight-forward age/growth study of these two species. Standard protocol was followed.

Validity of the findings

With the exceptions listed above, this study is important in establishing the basic aging and growth parameters of these two species when viewed as preliminary data for the possible future management of these fishes. Because the occurrence of these species is obviously rare in both the recreational and commercial catches in the area investigated, having size and longevity information for them can impact their treatment (small species complex vs. large species complex) in management decisions.

Additional comments

The layout of the paper might be reconsidered. In addition to editorial remarks made on the scanned manuscript, there are a few comments worthy of mention here.
Line 27 and line 48 - Please define here your geographic limits. Does SEUS include the GOM?
Line 46 - Should common names also be capitalized? AFS new standards?
Lines 60 - 78 - Consider placing this information in the results?
Lines 80 - 90 - Consider placing this information in the discussion?
Line 109 - Indicate again here geographically where the SEUS occurs
Line 118 - You indicated in the abstract that Total length = TL. It should also be introduced here.
Line 134 - Confused by "... generally recorded ..."
Line 175 - Does south Florida waters include the GOM, or does it only include the portion of southwest Florida that happens to be in that body of water?
Line 247, 248 - Italicize species names
Line 262 - Citation error
Line 270 - 274 - Are you indicating that the Dog Snapper was taken from deep waters off of NC vs, shallower water in more southerly areas? This is not clearly stated here.
Line 291, 292 - Citation errors
Line 332 - Citation not found in text
Line 335 - Poor literature listing
Line 369 - Isn't there a later edition of the FAO work?
Lines 405, 410 - Author's initials confused?
Line 532 - Explain what 'edge type' refers to, or cite the text for explanation.

Annotated reviews are not available for download in order to protect the identity of reviewers who chose to remain anonymous.

---

## Round 0.2 · accepted · Accept

Thank you very much for improving your manuscript and congratulations.